# ONESEARCH: THE UNIFIED END-TO-END GENERATIVE FRAMEWORK FOR E-COMMERCE SEARCH

## ABSTRACT

Traditional e-commerce search systems employ multi-stage cascading architectures (MCA) that progressively filter items through recall, pre-ranking, and ranking stages. While effective at balancing computational efficiency with business conversion, these systems suffer from fragmented computation and optimization objective collisions across stages, which ultimately limit their performance ceiling. To address these, we propose **OneSearch**, the first industrial-deployed end-to-end generative framework for e-commerce search. This framework introduces three innovations: (1) a Keyword-enhanced Hierarchical Quantization Encoding (KHQE) module, to preserve both hierarchical semantics and distinctive item attributes while maintaining strong query-item relevance constraints; (2) a multi-view user behavior sequence injection strategy that constructs behavior-driven user IDs and incorporates both explicit short-term and implicit long-term sequences to model user preferences comprehensively; and (3) a Preference-Aware Reward System (PARS) featuring multi-stage supervised fine-tuning and adaptive reward-weighted ranking to capture fine-grained user preferences. Extensive offline evaluations on large-scale industry datasets demonstrate OneSearch's superior performance for high-quality recall and ranking. The online A/B tests confirm its ability to enhance relevance in the same exposure position, achieving statistically significant improvements: +1.67% item CTR, +2.40% buyer, and +3.22% order volume. Furthermore, OneSearch reduces operational expenditure by 75.40% and improves Model FLOPs Utilization from 3.26% to 27.32%. The system has been successfully deployed across multiple search scenarios in TEST, serving millions of users, generating tens of millions of PVs daily.

## 1 INTRODUCTION

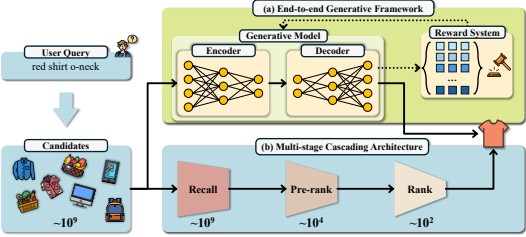

Figure 1: (a) Our proposed End-to-End generative retrieval framework, (b) the traditional multi-stage cascading architecture in E-commerce search.

E-commerce search systems aims to identify items satisfying both semantic and personalized criteria from hundreds of millions of candidates within one second. Traditional systems employ Multi-stage Cascading Architecture (MCA), which progressively filters items through recall ($10^9$ candidates), pre-ranking ($10^4$ candidates), and ranking ($10^2$ candidates) stages, as shown in Figure 1(b).

While MCA effectively balances response time and accuracy, it suffers from two fundamental limitations (Dubey et al., 2024a; Deng et al., 2025; Wei et al., 2025). First, fragmented compute: most serving resources are allocated to communication and storage rather than computation. Second, objective collision: heterogeneous optimization objectives across stages limit performance ceiling. Recall and pre-ranking stages prioritize coverage with lightweight models, while ranking emphasizes user preference reasoning with complex features. This multi-layer filtering risks discarding relevant items early, preventing optimal results regardless of subsequent model accuracy.

Recent efforts address these issues through either intra-stage optimization (Huang et al., 2020; Wang et al., 2021; Huang et al., 2013; Zhou et al., 2018; Guo et al., 2017) or cross-stage consistency (Zhang et al., 2023; 2025; Evnine et al., 2024), yet remain constrained by MCA's inherent limitations. The emerging generative retrieval (GR) paradigm offers a promising alternative by transforming matching-based frameworks into generation-based approaches (Rajput et al., 2023; Zheng et al., 2024; Pang et al., 2025; Deng et al., 2025; Guo et al., 2025; Zheng et al., 2025; Wei et al., 2025), eliminating multi-stage filtering through direct item generation.

However, e-commerce search presents unique challenges for GR adoption: (1) **Noisy item information**: sellers add irrelevant terms for exposure, creating lengthy descriptions with weak semantic order that mislead representation models; (2) **Strict relevance constraints**: queries typically contain 2-3 keywords where any attribute mismatch causes relevance issues—while semantic IDs provide hierarchical representations, they inevitably lose core attributes by prioritizing shared information; (3) **Latent intent inference**: uncovering user search intent from concise queries requires effectively combining query content with behavior profiles. To address these challenges, we propose **OneSearch**, an end-to-end generative framework for e-commerce search, which includes:

1) **K**eyword-enhanced **H**ierarchical **Q**uantization **E**ncoding module. KHQE employs keyword-enhanced semantic collaborative encoding to highlight core item attributes, using RQ-Kmeans for hierarchical feature encoding and OPQ for unique feature quantization, further reducing noise and then enhancing query-item relevance.

2) **M**ulti-view **U**ser Behavior **Seq**uence (Mu-Seq) injection strategy. This strategy constructs behavior-driven user IDs with weighted decay sequences, explicitly incorporates short sequences for recent preferences, and implicitly models long sequences for comprehensive user profiles.

3) **P**reference **A**ware **R**eward **S**ystem (PARS). We implement multi-stage supervised fine-tuning (SFT) for semantic alignment and personalization, followed by adaptive reward-weighted ranking combining hierarchical behavior signals with list-wise preference optimization.

Extensive evaluations demonstrate OneSearch's superiority. Online A/B tests show statistically significant improvements: +1.67% item CTR (Click Through Rate), +2.40% buyer volume, and +3.22% order volume, while reducing operational expenditure by 75.40% and improving Model FLOPs Utilization from 3.26% to 27.32%. OneSearch is successfully deployed across multiple TEST search scenarios, serving millions of users with tens of millions of daily Page Views (PVs).

The main contributions of this work are summarized as follows:

- A keyword-enhanced hierarchical quantization encoding balancing context features and collaborative signals while strengthening relevance constraints.
- A multi-view behavior sequence injection strategy, integrating user behavior sequences into ID representations and leveraging explicit/implicit prompts to enhance GRs' reasoning about user profiles and preferences.
- A preference aware reward system with multi-stage SFT and adaptive reward modeling for personalized ranking capability.
- Finally, we present OneSearch, the first industrial-deployed end-to-end generative framework for e-commerce search, validated through comprehensive offline and online experiments.

## 2 METHODOLOGY

This section details OneSearch, our end-to-end e-commerce search framework, in four parts.

### 2.1 KEYWORD-ENHANCED HIERARCHICAL QUANTIZATION ENCODING

Encoding items into Semantic IDs (SIDs) is crucial for generative retrieval models. This process converts continuous semantic representations into discrete ID sequences using coarse-to-fine quantization, ensuring items with the same SID share same information(Rajput et al., 2023; Deng et al., 2025; Ju et al., 2025). However, common quantization methods tend to tokenize shared signals using fixed vocabulary, losing distinctive attributes. We propose KHQE combining domain knowledge extraction, RQ-Kmeans for hierarchical encoding, and OPQ for unique feature quantization.

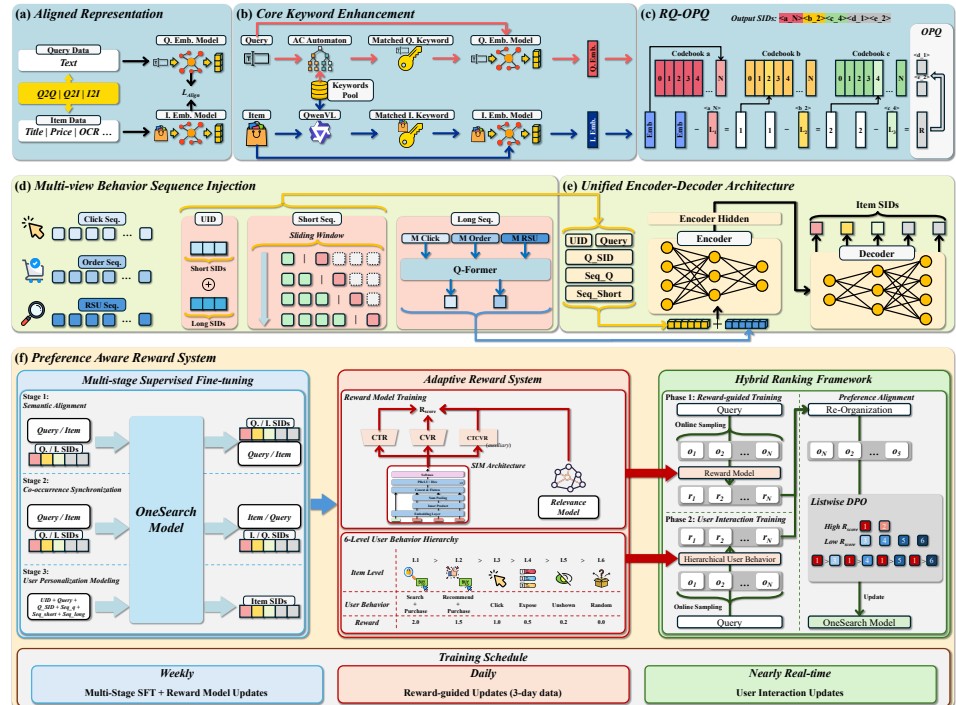

Figure 2: The Framework of OneSearch includes: 1) Keyword-enhanced Hierarchical Quantization Encoding, which adopts aligned representation and core keyword to construct a hierarchical quantization tokenization schema; 2) Multi-view Behavior Sequence Injection, utilizing sequences to reason the user profiles and preferences; 3) Unified encoder-decoder Architecture that integrates produced features for generative retrieval; and 4) Preference Aware Reward System, containing a multi-stage SFT procedure and an adaptive reward system to enhance the ranking capability.

**Aligned collaborative and semantic representation.** We integrate semantic knowledge with collaborative signals by aligning the representations of historically interactive query-item pairs. Firstly, we select high-quality query2query, item2item, and query2item pairs from real user search logs using existing retrieval models like ItemCF (Sarwar et al., 2001) and Swing (Yang et al., 2020). Then we collect the content information like query text, item title, item price, keywords, OCR (image-to-text), as well as the statistical business characteristics, such as the number of clicks, add-to-cart, and purchases during a certain time. All these features are processed with a distilled BGE (Xiao et al., 2023) to generate a content embedding for each query $e_q$ and item $e_i$. Finally, we filter pairs with cosine similarity larger than 0.6 to ensure content relevance.

We design four interrelated tasks to align collaborative and semantic representation: 1) the query2query and item2item contrastive loss $\mathcal{L}_{q2q}$, $\mathcal{L}_{i2i}$ to align representations of collaboratively similar pairs, 2) a query2item contrastive loss $\mathcal{L}_{q2i}$ to ensure that BGE can reflect real business characteristics, 3) a query2item margin loss $\mathcal{L}_{rank}$ to further learn the collaborative signal deviation of query-item pairs with different behavior levels, 4) a hard sample relevance correction loss $\mathcal{L}_{rel}$. Then we train the aligned model by the total loss with adjustable parameters $\lambda_i$ as:

$$\mathcal{L}_{align} = \lambda_1 \cdot \mathcal{L}_{q2q} + \lambda_2 \cdot \mathcal{L}_{i2i} + \lambda_3 \cdot \mathcal{L}_{q2i} + \lambda_4 \cdot \mathcal{L}_{rank} + \lambda_5 \cdot \mathcal{L}_{rel}, \quad (1)$$

**Core Keyword Enhancement.** Item textual information often contain redundant, irrelevant, or conflicting attributes. Although boosting exposure, these disordered attributes makes it difficult for encoders to model key information. Here we propose using core keyword features to enhance textual representation, obtaining keyword-dominated semantic IDs.

Using NER (detailed shown in Appendix A.3), we extract 18 structured attributes and mine recent query-item pairs as labeled data. Core keywords are selected from PV-ranked attribute lists. Qwen-VL (Bai et al., 2023) identifies item keywords, while Aho-Corasick Automaton (Aho & Corasick, 1975) matches query keywords during inference. All these core keywords are fed into the trained model to obtain vectors $e_k^i$ consistent with the item representation distribution. The final optimized

representations for each query $e_q^o$ and item $e_i^o$ are given by:

$$e_q^o = \frac{1}{2}(e_q + \frac{1}{m}\sum_{i=1}^{m} e_k^i), \quad e_i^o = \frac{1}{2}(e_i + \frac{1}{n}\sum_{j=1}^{n} e_k^j). \tag{2}$$

**RQ-OPQ Hierarchical Quantization Tokenization.** Common SID tokenizers like VQ-VAE (van den Oord et al., 2018), RQ-VAE (Lee et al., 2022), and RQ-Kmeans (Luo et al., 2024) prioritize encoding shared item features at the expense of distinctive characteristics, limiting generative GR performance. To address this limitation, we introduce a hybrid tokenizer called RQ-OPQ. We first employ RQ-Kmeans as the foundational tokenizer to capture hierarchical semantic structure. Then, we uniquely quantize the residual embedding discarded after RQ-Kmeans' final clustering step using OPQ. This approach allows RQ-Kmeans to model hierarchical relationships while OPQ simultaneously captures complementary lateral features. The resulting RQ-OPQ tokenizer provides a more comprehensive representation of fine-grained item characteristics, significantly strengthening the relevance constraints for downstream GR models. (Implementation specifics, codebook architectures, performance metrics, and ablation studies are detailed in the Appendix A.3).

## 2.2 MULTI-VIEW BEHAVIOR SEQUENCE INJECTION

**Behavior Sequence Constructed User IDs.** Random Hash IDs in Tiger (Rajput et al., 2023) do not adequately represent user personalization. Here, we propose a behavior sequence-constructed user ID for distinctive representation. Formally, the short behavior sequence consists of latest clicked items, denoted as $Seq_{short}$, length is $m$, and the long behavior sequence contains chronologically clicked items, denoted as $Seq_{long}$, length is $n$. The user ID $uid$ is computed as the concatenation of $SID_{short}$ and $SID_{long}$:

$$SID_{\text{short/long}} = \left\lceil \sum_{k \in \mathcal{K}} \gamma_k \cdot SID_k \right\rceil, \ \gamma_k = \frac{e^{\sqrt{k}}}{\sum_{i \in \mathcal{K}} e^{\sqrt{k}}} \tag{3}$$

where $K = \{s_1, s_2, \ldots, s_m\}/\{l_1, l_2, \ldots, l_n\}$ denote the short/long behavior sequence. For cold-start users, we count the most clicked items for each query based on query-item occurrence and sort them by page views as default sequences.

**Explicit Short Behavior Sequence.** Short sequences primarily reflect recent preferences while long sequences represent user profiles. Therefore, for generative retrieval, explicitly inputting short sequences makes it easier to predict likely click categories. In e-commerce search, the short behavior sequences include latest queries $Seq_q$ and clicked items $Seq_{short}$. We input their SIDs into the prompt, following the constructed user ID and input query.

**Implicit Long Behavior Sequence.** Long sequences consist of click, order, and relevant search unit (RSU) sequences(Guo et al., 2023), with lengths of up to $10^3$, making direct prompt integration infeasible. For each item, we map its keyword-enhanced embedding $e_i^o$ to a corresponding semantic ID, and get RQ clustering centroid representation through lookup. We aggregate centroids by levels, enabling systematic preference learning while saving resources. Each item in the long-term historical sequence is replaced by the features of its RQ clustering centroid representative:

$$\text{Item}_{sid} = \text{RQ}(e_i^o), \quad \text{Item}_{emb} = \text{EmbLookUp}(\text{Item}_{sid}) \tag{4}$$

For long-term sequence, overall behavior embedding is shown as:

$$\mathbf{M}_{click/order/RSU} = \left\{ \sum_{i=1}^{L1} \text{Item}_{emb}^1, \sum_{i=1}^{L2} \text{Item}_{emb}^2, \sum_{i=1}^{L3} \text{Item}_{emb}^3 \right\}, \tag{5}$$

$$\mathbf{Q}^{(i)} = \text{QFormer}(\mathbf{M}_{click}, \mathbf{M}_{order}, \mathbf{M}_{RSU}),$$

where $\mathbf{M}_{click}$, $\mathbf{M}_{order}$, and $\mathbf{M}_{RSU}$ are click / order / RSU sequence item embeddings, and share the same size $\mathbf{M} \in \mathbb{R}^{N_M \times d_{model}}$ ($d_{model} = 768$).

## 2.3 UNIFIED ENCODER-DECODER ARCHITECTURE

The input of OneSearch contains: 1) Distinctive user ID $uid$. 2) Entered query $q$ and its $SID_q$; 3) User short behavior sequence, containing the historical search queries $Seq_q = \{q_1, q_2, \ldots, q_n\}$, the short clicked item sequence $Seq_{short} = \{s_1, s_2, \ldots, s_n\}$; 4) Implicit long behavior sequence $Seq_{long}^{emb}$; 5) User profile information $\mathcal{U}$, which is the crowd portrait fitted by the platform. OneSearch

directly outputs corresponding item lists $\mathcal{I}$. The model adopts either encoder-decoder models (e.g. BART (Lewis et al., 2019), mT5 (Xue et al., 2020)), or the decoder-only models (e.g. Qwen3 (Yang et al., 2025)) as the backbone $\mathcal{M}$. The inference flow can be formalized as:

$$\mathcal{I} := \mathcal{M}(uid, q, SID_q, Seq_q, Seq_{short}, Seq_{long}^{emb}, \mathcal{U}). \tag{6}$$

As illustrated in Figure 2, our model adheres to the transformer-based (Vaswani et al., 2017) architecture, comprising an encoder that models $\langle user, query, seq \rangle$ information, and a decoder dedicated to item generation. We adopt encoder-decoder models for deployment due to architecturally accelerated training and inference. For unified training, we insert $t_{[\text{BOS}]}$ and $t_{[\text{EOS}]}$ at boundaries, with $t_{[\text{SEP}]}$ between adjacent elements. The inference output of $\mathcal{M}$ is the SIDs, which can be adjusted by constrained or unconstrained beam search. While constrained search guides output to valid SIDs, it increases decoding complexity, and unconstrained explores all sequences without explicit rules.

## 2.4 Preference Aware Reward System

Compared to sequence coherence in recommendations, strong relevance constraints between queries and items in search pose greater challenges. For GR models, we must achieve semantic alignment between SIDs and text descriptions while directly generating items meeting relevance constraints and user preferences. We propose PARS with multi-stage supervised fine-tuning and adaptive reward system for personalized ranking capability. The overall training framework is depicted in Figure 2(f).

**Multi-stage Supervised Fine-tuning.** Since basic architectures (e.g., BART, T5) are pretrained on text corpus while OneSearch uses SID representations, we first achieve semantic alignment then instruct generation of user-aligned items through three stages:

1. **Semantic Content Alignment**: Three sub-tasks: (a) Query/Item Text→SID generation, (b) SID→text reconstruction, (c) Text/item→category prediction. First two tasks align SID and text content,while the category prediction ensures relevance.
2. **Co-occurrence Synchronization**: Mutual prediction between query↔item and query SID↔item SID. Without user characteristics, this stage learns intrinsic semantics and collaborative relationships from interactive corpus.
3. **User Personalization Modeling**: After the above stages, we introduce user information aligning with online inference. We concatenate user ID, query, $SID_q$, $Seq_q$, $Seq_{short}$, and $Seq_{long}^{Emb}$ as input with item SID as training label for distinctive personalization.

We apply sliding window data augmentation to short sequences to guide learning of user interest changes. The sliding window generates new segments with subsequent items as prediction targets by sliding along $Seq_{short}$ (Zhou et al., 2024). With maximum window length limitation, we augment $m$ samples for $Seq_{short} = \{s_1, s_2, \ldots, s_m\}$, helping handle new users with limited history.

**Adaptive Reward System.** Unlike OneRec-V1's (Zhou et al., 2025a) weighted P-Score with single reward model and Early Clipped GRPO, we use real online interactions as hierarchical feedback signals. We adopt adaptive-weighted rewards (Guo et al., 2025) to construct training data and implement user-behavior-guided hybrid ranking for personalized preferences.

Table 1: The preference-aware reward system combines a three-stage fine-tuning process (semantic alignment, co-occurrence, personalization) and an adaptive ranking mechanism.

| Procedure | SFT Stage 1 | SFT Stage 2 | SFT Stage 3 | | RL Stage | |
|---|---|---|---|---|---|---|
| **Objective** | Semantic alignment | $\langle q, i \rangle$ co-occurrence | User personalization | | Preference Alignment | |
| **Component** | query $\leftrightarrow$ SID 
 item $\leftrightarrow$ SID 
 query/item $\mapsto$ category 
 SID $\mapsto$ category | query $\leftrightarrow$ item 
 query_SID $\leftrightarrow$ item_SID | $\begin{bmatrix} uid \,\&\, q \\ SID_q \,\&\, Seq_q \\ Seq_{short} \\ Seq_{long}^{emb} \end{bmatrix}$ | $\mapsto$ item_SID | $\begin{bmatrix} \text{user \& query} \\ \text{seq. feat.} \\ item_{win} \\ item_{lose} \end{bmatrix}$ | $\mapsto$ Rank Score |

**Adaptive-weighted Reward Signal.** Following OneSug (Guo et al., 2025) we categorize interactive behaviors into six levels: (1) purchased in search, (2) same-category purchased in recommendation, (3) clicked, (4) exposed-not-clicked, (5) unshown same-category, (6) random other-category. Base weights are $\lambda = [2.0, 1.5, 1.0, 0.5, 0.2, 0.0]$. Considering items with higher recent CTR/CVR (Conversion Rate) are more likely selected, we utilize these two metrics to construct adaptive-weighted rewards. However, CTR and CVR often suffer from biased estimation. For example, a newly released item that was exposed only once and then clicked would have CTR at 100%. Conversely,

genuinely popular items are often exposed by online MCA under various similar but suboptimal queries, resulting in lower CTR and CVR. Therefore, we calibrate these two metrics as follows:

$$Cnt_T = \log((Cnt_{pos} + 10) \cdot (Cnt_{clk} + 10) \cdot (Cnt_{order} + 10)).$$

$$Ctr_i = \frac{\log(Cnt_{clk} + 10)}{Cnt_T}, \quad Cvr_i = \frac{\log(Cnt_{order} + 10)}{\log(Cnt_{clk} + 10)}. \tag{7}$$

The weighted reward score is then defined as:

$$r(q, i) = 2\lambda \cdot \frac{Ctr_i \cdot Cvr_i}{Ctr_i + Cvr_i}. \tag{8}$$

For each positive sample $i_{pos}$ and negative sample $i_{neg}$, the user preference difference $rw_\Delta$ is:

$$rw_\Delta = 1.0 \, / \, [r(q, i_{pos}) - r(q, i_{neg})], \tag{9}$$

where smaller $rw_\Delta$ encourage the model to distinguish nuanced differences in user behaviors.

**Reward Model Training.** As discussed in OneRec-V2(Zhou et al., 2025b), the reward model in OneRec-V1 employs restricted sampling from a small subset of users to approximate global behavior, potentially learning specific patterns or biases that do not yield actual improvements. However, we also diverge from the feedback-driven preference alignment proposed in OneRec-V2, as the adoption of GRPO and its variants (e.g., ECPO, GBPO) tends to introduce more irrelevant SIDs, and preference rewards require careful tuning for e-commerce search. Here we design an intuitive three-tower SIM (Qi et al., 2020)) architecture, with each tower learning CTR, CVR, CTCVR (Ma et al., 2018) using binary cross-entropy. The preference score is:

$$Rscore = \lambda_1 \cdot CTR + \lambda_2 \cdot CVR + \lambda_3 \cdot CTCVR + 10 \cdot \lambda_4 \cdot S_{Rel}, \tag{10}$$

where $\lambda_i$ represents tuned weights (set to 1 in our experiments). To ensure that results generated by OneSearch meet relevance constraints, we additionally incorporate an offline-calculated relevance score $S_{Rel}$ with an amplified weight ($10 \cdot \lambda_4$).

This reward model differs from the click prediction model in the ranking stage of online MCA in two key aspects: (1) Feature dimensionality: While the ranking model utilizes thousands of features, our reward model only takes user ID, query, user behavior sequence, and user profile as input, matching OneSearch's input space. (2) Sampling strategy: We additionally include items from the same category clicked in recommendation scenarios as training samples, with labels (1,1,1) for purchased items and (1,0,0) for clicked items. For computational efficiency, the reward model directly leverage the online MCA ranking model, as we only distill the ranking order rather than absolute scores.

**Hybird Ranking Framework.** We employ two-phase alignment. First, we collect real queries and use reward model to rerank OneSearch outputs, selecting samples with ranking changes for list-wise DPO training. Clicked or advanced items serve as positives; pushed-back items as negatives. The optimization objective is:

$$\mathcal{L} = -\mathbb{E}\left[\log \sigma\left(\log \sum_{i_l \in \mathcal{I}_l} \exp\left(rw_\Delta \max\left(0, \hat{r}_\theta(x_u, i_w) - \hat{r}_\theta(x_u, i_l) - \delta\right)\right)\right) + \alpha \log \pi_\theta(i_w | x_u)\right], \tag{11}$$

where $\mathcal{I}_l$ denotes the set of negative samples, and $\hat{r}_\theta(x_u, i_w)$ and $\hat{r}_\theta(x_u, i_l)$ represent rewards implicitly defined by the language model $\pi_\theta$ and reference model $\pi_{\text{ref}}$:

$$\hat{r}_\theta(x_u, i_{w/l}) = \beta \log \frac{\pi_\theta(i_{w/l} | x_u)}{\pi_{\text{ref}}(i_{w/l} | x_u)}. \tag{12}$$

The term $\log \pi_\theta(i_w | x_u)$ represents the log-likelihood (NLL loss) from the SFT stage.

Noted that by combining the list-wise preference alignment with log-likelihood prediction of preferred samples, we establish a novel hybrid paradigm for generative ranking. Since the reward model trains on MCA data, it inherently limits OneSearch's performance. Thus, phase two uses pure user interactions: positives from top behaviors (purchased, same-category purchased, clicked) and negatives from bottom behaviors (exposed-not-clicked, unshown same-category, random), with the same loss. In practice, we periodically perform first-phase RL with reward model-generated samples to ensure online distribution adherence and learn from the MCA ranking model's thousands of features. The second phase updates near-streaming with user interaction data to overcome distribution limitations and fully leverage generative inference capabilities. This dual approach enables OneSearch to benefit from both MCA's rich feature space and direct user feedback.

Table 2: Offline performances of our proposed method with onlineMCA on the industry dataset. The best results are in bold, and sub-optimal results are underlined in each column. The "w/o ranking" means "without ranking", and the "\+ keywords " means "add keywords optimizations"

| Method | order (30k) | | click (30k) | |
|---|---|---|---|---|
| | **HR@350** | **MRR@350** | **HR@350** | **MRR@350** |
| OnlineMCA | 51.74% | 19.26% | 64.40% | 16.89% |
| w/o ranking | 75.75% | 4.19% | 80.23% | 3.00% |
| OPQ (8/256) | 19.43% | 9.55% | 22.57% | 7.42% |
| (1024-1024-1024) | 57.39% | 9.12% | 63.63% | 7.46% |
| (2048-1024-512) | 58.29% | 10.79% | 65.39% | 8.86% |
| (4096-1024-256) | 58.57% | 11.21% | 64.51% | 9.24% |
| (4096-1024-512) | 59.58% | 14.29% | 62.49% | 11.82% |
| \+ keywords | 62.38% | 14.30% | 66.14% | 12.10% |
| \+ l3 balanced | 63.16% | 13.59% | 68.26% | 11.67% |
| \+ Adaptive RS | 64.33% | 16.11% | 68.94% | 13.80% |
| RQ-OPQ (2/256) | 65.05% | 15.33% | 68.88% | 12.90% |
| \+ Adaptive RS | **66.46%** | **18.38%** | **71.06%** | **16.33%** |

## 3 EXPERIMENT

In this section, we conduct comprehensive evaluations on practical industry datasets and online A/B tests to verify the feasibility of OneSearch, followed by ablation studies.

**Datasets.** We extract the user interactive pairs from TEST's mall search platform between May 2025 and August 2025 to facilitate the supervised fine-tuning and DPO. It contains about 1 billion PVs, all the offline and ablation experiments were conducted on the full or part of this data.

**Evaluation Metrics.** We take into account the recall and ranking performance. We employ HitRate (HR) and Mean Reciprocal Ranking (MRR) for recall and ranking performance, which are standard metrics in search and recommendation systems.

**Baselines.** Unlike simulation approaches, we use the actual production system with multiple recall mechanisms and complex ranking with thousands of features and compare against the real onlineMCA. Details are in Appendix A.4.

### 3.1 OFFLINE PERFORMANCE

We evaluated 30,000 click pairs and 30,000 order pairs from user search logs, computing HR@350 and MRR@350. Table 2 shows that pre-ranking prioritizes recall over precision (75.75% HR but 4.19% MRR for orders), while ranking emphasizes intent positioning. This demonstrates MCA's optimization objective collision—final ranking is constrained by pre-ranking outputs.

Testing various RQ-Kmeans and KHQE configurations (Table 6), we found higher codebook utilization rate (CUR) and independent coding rate (ICR) improve performance. Core keyword enhancement and L3 balanced k-means both provide improvements. Besides, adaptive reward preference learning significantly enhances ranking (+1.80% HR@350, +3.24% MRR@350 average).

Our final configuration, RQ-OPQ (2/256) with Adaptive RS, combines RQ-Kmeans (4096-1024-512) for hierarchical encoding with OPQ (256-256) for residual quantization, trained using the full preference aware reward system. This achieves superior recall (66.46% vs. 51.74% for orders) and comparable MRR (18.38% vs. 19.26%) to onlineMCA, effectively balancing personalized ranking with intent-matching. The configuration maintains robust performance across both click and order metrics, validating our hybrid tokenization approach. This configuration would be called OneSearch in the following section for brevity. Additional implementation details are in Appendix A.4.

### 3.2 ABLATION STUDY

**Multi-view Behavior Sequences.** Table 3 demonstrates each component's contribution. Sequence-constructed user IDs outperform hashing IDs (+1.33% HR@350). Short sequences provide the

Table 3: Ablation study of multi-view behavior sequence injection. Slid. Window means the sliding window strategy.

| Method | order (30k) | | click (30k) | |
|---|---|---|---|---|
| | HR@350 | MRR@350 | HR@350 | MRR@350 |
| OneSearch | 66.46% | 18.38% | 71.06% | 16.33% |
| w/o User SIDs | -0.94% | -0.37% | -1.72% | -0.36% |
| w/o $Seq_{short}$ | -3.43% | -1.53% | -4.15% | -1.32% |
| w/o $Seq_{long}^{emb}$ | -2.26% | -1.01% | -3.00% | -1.05% |
| w/o Slid.Window | -1.95% | -0.81% | -1.80% | -0.70% |

largest gains (+3.79% HR@350, +1.43% MRR@350), validating explicit preference modeling. Long sequences and sliding window augmentation further enhance performance.

**Tokenization Stability.** E-commerce inventory changes constantly, especially during shopping festivals, potentially disrupting pre-calculated SID pools. We tested tokenizer stability by constructing RQ-Kmeans and RQ-OPQ using July 15 items and tracking performance through August 18 promotions. Figure 3 shows minimal degradation: RQ-Kmeans decreased 1.11% in CUR while RQ-OPQ only 0.43%, validating RQ-OPQ's superior robustness to inventory changes. Detailed RQ-OPQ ablations are in Appendix A.5.

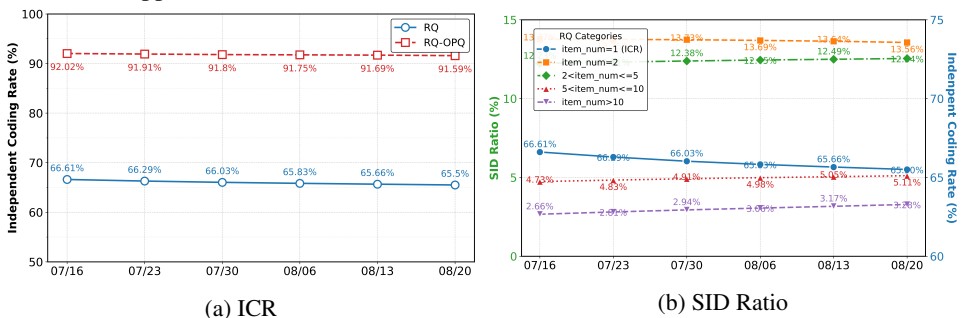

(a) ICR  (b) SID Ratio

Figure 3: The ICR and SID ratio indicators of tokenizations after regular time intervals.

### 3.3 ONLINE A/B TESTING

Table 4: Online results for A/B testing. The black fonts indicate that the statistical significance (P-value) is smaller than 0.05, while the gray ones are larger than 0.05 indicating low confidence.

| Method | Item CTR | PV CTR | PV CVR | Buyer | Order |
|---|---|---|---|---|---|
| MCA w/o ranking | -9.97% | -20.33% | -11.55% | -28.78% | -39.14% |
| OneSearch[1] | -1.10% | -2.06% | +0.39% | +1.27% | -2.22% |
| OneSearch[1]$_{RM}$ | +1.40% | +3.05% | +1.94% | +1.92% | +1.59% |
| OneSearch[2] | +1.45% | +1.40% | -0.12% | -0.58% | -0.69% |
| OneSearch[2]$_{RM}$ | **+1.67%** | **+3.14%** | **+1.78%** | **+2.40%** | **+3.22%** |

We conducted rigorous A/B tests on TEST's mall search platform. Table 4 presents results for two OneSearch versions: OneSearch[1] (RQ-Kmeans without long sequences) and OneSearch[2] (full optimizations), each tested with and without reward model reranking ($RM$).

Table 4 demonstrates that the base generative model achieves comparable performance to the complex MCA system. With RQ-OPQ and long behavior sequences, OneSearch[2] improves item CTR by 1.45% and PV CTR by 1.40%. Incorporating reward model selection (OneSearch[2]$_{RM}$) yields statistically significant gains across all metrics: +1.67% item CTR, +3.14% PV CTR, +1.78% PV CVR, +2.40% buyer volume, and +3.22% order volume.

For a clearer comparison, we perform additional experiments on the online search system, named MCA w/o ranking, which only uses the "recall and pre-ranking" module to predict the items, without the ranking stage. It significantly reduces all indicators, especially with 28.78% in Buyer, 39.14% in Order volume. This indirectly verifies that OneSearch has comparable ranking capabilities. These

outstanding results show that OneSearch outperforms the onlineMCA and indicate it can update the complicated online system to a more balanced state without generating seesaw effects. Last but not least, we also observe improvements in manual evaluations, detailed manual evaluation results are provided in Appendix A.6.

Ultimately, OneSearch has been successfully deployed for the entire traffic on the e-commerce detail page search engine in TEST, 50% traffic on the mall search, and 20% traffic on the homepage e-commerce search platform for further investigation, which serves millions of users generating tens of millions of PVs daily.

### 3.4 FURTHER ANALYSIS

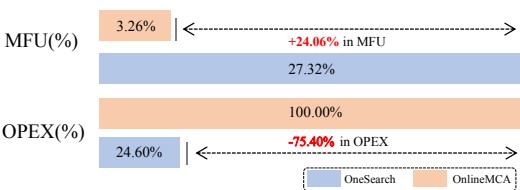

Figure 4: The comparisons of MFU and OPEX for onlineMCA and OneSearch.

**Computational Efficiency.** We measured Model FLOPs Utilization (MFU) on flagship GPUs during serving inference. As shown in Figure 4, OneSearch achieves 27.32% MFU compared to onlineMCA's 3.26%, an 8.4× improvement. This significantly outperforms traditional MCA systems while approaching the 40% MFU typical of LLMs on H100 GPUs (Dubey et al., 2024b). Furthermore, OneSearch reduces operational expenditure (OPEX) to 24.60% of the online search pipeline by eliminating communication and memory overhead between stages. Additional discussion is provided in the Appendix A.7.

## 4 RELATED WORKS

In recent years, Generative Retrieval (GR) has garnered attention from both academia and industry due to its remarkable performance. Notable contributions in this area include Tiger (Rajput et al., 2023), DSI (Tay et al., 2022), and LC-REC (Zheng et al., 2024). Most GR models serve merely as supplementary recall sources within online systems, thereby overlooking these models' inherent rich semantic and powerful reasoning abilities for potential use in (pre-)ranking stages. In the area of video recommendation, OneRec (Deng et al., 2025) was the first to unify recall, pre-ranking, and ranking within a single generative model. Most advancements in generative retrieval have been focused on recommendations. This is because search systems face three major challenges: 1) multiple and low-density item information, 2) strong relevance constraints between search queries and items, and 3) inference barriers to users' potential search intentions. Consequently, the current traditional e-commerce search systems still adopt a multi-stage cascading architecture. However, some efforts have been made to optimize current search systems using GR, e.g., GenR-PO (Li et al., 2024), GRAM (Pang et al., 2025) and OneSug (Guo et al., 2025). These GR methods demonstrate appealing performance in the realm of search, recommendation, bottom navigation, advertising, and query suggestion. Inspired by these works, we proposed OneSearch, which is suitable for e-commerce search, to achieve open-set input to closed-set output. See extended discussion in Appendix A.2.

## 5 CONCLUSION

In this paper, we present OneSearch, a pioneering end-to-end generative framework for e-commerce search system that effectively overcomes the limitations of traditional multi-stage cascading architecture. By employing a unified generative model, introducing the keyword-enhanced hierarchical quantization encoding, multi-view behavior sequences injection, and preference aware reward system, OneSearch achieves superior semantic understanding and personalization modeling. The preference-aware reward strategy further refines the model's ability to capture user preferences, leading to improved ranking performance. Extensive offline and online evaluations confirm OneSearch's effectiveness in boosting click-through rates and business conversions. Its successful deployment on multiple TEST search scenes underscores its practical applicability and potential to enhance industry revenue. OneSearch sets a new benchmark for industrial search solutions, paving the way for future advancements in generative retrieval methods.

ETHICS STATEMENT

This research uses anonymized user interaction data from an e-commerce platform in compliance with privacy policies and data protection regulations. The deployment affects millions of users, and we have implemented monitoring systems to detect potential biases or adverse effects. A/B testing methodology ensures gradual rollout to minimize user risks. We recognize algorithmic search systems can influence user behavior and have taken measures to promote relevant, diverse results across product categories. The research balances commercial objectives with user value, and the system continues to be monitored for ethical implications.

REPRODUCIBILITY STATEMENT

To support reproducibility and advance future research, we will release the complete OneSearch codebase, pre-trained model weights, and training scripts upon publication. Comprehensive implementation details, hyperparameters, and experimental configurations are provided in Appendix A.4. We will also provide detailed documentation of our data preprocessing pipeline, model architecture specifications, and evaluation protocols to enable researchers to replicate our methodology and extend our work to new domains and datasets.

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

# A APPENDIX

## A.1 USE OF LARGE LANGUAGE MODELS

We hereby declare that throughout the innovative conceptualization, code development, data construction, manuscript preparation, and creation of all figures and tables in this research, **no artificial intelligence generation tools were utilized**.

## A.2 RELATED WORKS

**Generative Retrieval and Recommendation.** In recent years, Generative Retrieval (GR) has garnered significant attention from both academia and industry due to its remarkable performance. This emerging retrieval paradigm, which regards large-scale retrieval as sequence-to-sequence generation tasks, has outperformed traditional ANN-based models such as EBR (Huang et al., 2020) and RocketQA (Qu et al., 2021), spurring increased exploration in the fields of search and recommendation. Notable contributions in this area include Tiger (Rajput et al., 2023), DSI (Tay et al., 2022), and LC-REC (Zheng et al., 2024). Tiger (Rajput et al., 2023) pioneered the development of end-to-end generative retrieval models for sequential recommendation, introducing semantic IDs (SID) derived

from each item's content information for efficient item representation. LC-REC (Zheng et al., 2024) proposed adapting large language models (LLMs) by integrating collaborative semantics for recommendation, utilizing a series of specially designed tuning tasks.

Most GR models serve merely as supplementary recall sources within online systems, thereby overlooking these models' inherent rich semantic and powerful reasoning abilities for potential use in (pre-)ranking stages. In the area of video recommendation, OneRec (Deng et al., 2025) was the first to unify recall, pre-ranking, and ranking within a single generative model. This was achieved with the assistance of session-wise generation and iterative preference alignment, resulting in substantial improvements in practical online metrics. EGA (Zheng et al., 2025) represents a significant departure from both traditional multi-stage cascading architectures (MCA) and existing generative retrieval models by introducing a unified framework that holistically models the entire advertising pipeline. UniROM (Qiu et al., 2025) employs a hybrid feature service to efficiently decouple user and advertising features, and RecFormer (Li et al., 2023), a variation of Transformer, captures both intra- and cross-sequence interactions.

**Generative Retrieval for Search.** These two years, most advancements in generative retrieval have been focused on recommendations. This is because search systems face three major challenges: 1) multiple and low-density item information, 2) strong relevance constraints between search queries and items, and 3) inference barriers to users' potential search intentions. Consequently, the current traditional e-commerce search systems still adopt a multi-stage cascading architecture. However, some efforts have been made to optimize current search systems using generative retrieval (GR).

The first example is GenR-PO (Li et al., 2024), which utilizes multi-span identifiers to represent raw item titles. This approach transforms the task of generating titles from queries into the task of generating multi-span identifiers from queries, thereby simplifying the generation process. Subsequently, a constrained search method is employed to identify key spans for retrieving the final item, which has proven beneficial for online recall systems. Another notable example is the Generative Retrieval and Alignment Model (GRAM) (Pang et al., 2025), which performs joint training on text information from both queries and products to generate shared text identifier codes. GRAM employs a co-alignment strategy to optimize these codes for maximizing retrieval efficiency and is deployed on the JD search engine to enhance both the recall and pre-ranking stages.

The OneSug (Guo et al., 2025) in query suggestion, which incorporates a prefix2query representation enhancement module to enrich prefixes using semantically and interactively related queries to bridge content and business characteristics, an encoder-decoder generative model that unifies the query suggestion process, and a reward-weighted ranking strategy with behavior-level weights to capture fine-grained user preferences. It is the first end-to-end generative framework for e-commerce query suggestion, and has been verified to have substantial improvements in user clicks and conversion. These GR methods demonstrate appealing performance in the realm of search, recom-

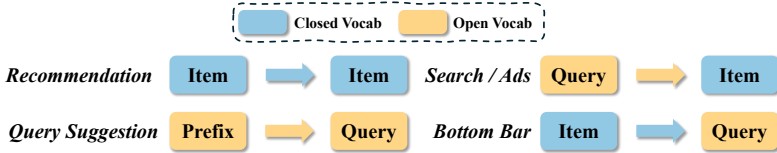

Figure 5: The input and output differences among Recommend, Search/Ads, Query Sug and Bottom Bar.

mendation, bottom navigation, advertising, and even query suggestion. They are not suitable for e-commerce search. As illustrated in Figure 5, the inputs and outputs of recommendation are the closed-vocabulary items or videos, thus the pure semantic ID tokenization is suitable for its diverse item generation. The inputs and outputs of query suggestion are the full open-vocabulary textual descriptions, so that it can directly use the transformer architecture. For the bottom bar and search engine, either the inputs or the outputs are open-vocabulary, which represents a significant departure from both OneRec and EGA.

### A.3 Keyword-enhanced Hierarchical Quantization Encoding Details

#### A.3.1 Core Keyword Enhancement Details

To enhance the role of core keywords in encoding, the core keyword enhancement scheme introduce 18 structured attributes (shown in Table 5) that improves the codebook utilization rate (CUR) of RQ-Kmeans at each level and further increases the independent coding rate (ICR). For example, with a configuration of 4096-1024-512 in Table 6, it results in a 0.10% CUR increment for Level 1, 24.84% for Level 2, and 26.15% for Level 3, as well as the overall ICR increasing by 6.86%.

Table 5: 18 structured attributes using NER in the TEST e-commerce search platform.

| Attribute Types | | | | | | | | |
|---|---|---|---|---|---|---|---|---|
| Entity Scene | Modifier Specifications | Brand Price | Material Model | Style Anchor | Function Series | Location Marketing | Audience Season | Color Pattern |

Table 6: The codebook utilization rate (CUR) and independent coding rate (ICR) for various RQ-Kmeans configurations. The last $^+$ means balanced operation for all levels.

| Configurations | $CUR_{L1}$ | $CUR_{L1*L2}$ | $CUR_{Total}$ | $ICR$ |
|---|---|---|---|---|
| 1024-1024-1024 | 100% | 54.27% | 1.72% | 36.67% |
| \+keywords | 100% | 65.40% | 2.03% | 40.25% |
| 2048-1024-512 | 100% | 46.88% | 1.98% | 37.80% |
| \+keywords | 100% | 57.16% | 2.51% | 40.76% |
| 4096-1024-256 | 99.90% | 39.21% | 2.27% | 36.98% |
| \+keywords | 100% | 48.95% | 2.94% | 40.52% |
| \+l3 balanced | 100% | 48.95% | 10.31% | 60.01% |
| 4096-1024-512 | 99.90% | 39.21% | 1.30% | 40.54% |
| \+keywords | 100% | 48.95% | 1.64% | 43.32% |
| \+l3 balanced | 100% | 48.95% | 7.03% | 68.08% |
| 4096-1024-512$^+$ | 99.93% | 41.45% | 0.51% | 33.47% |

Table 7: Performance comparisons of three tokenization schemas evaluated on the real click pairs.

| Method | $CUR_{Total}$ | $ICR$ | Recall@10 | MRR@10 |
|---|---|---|---|---|
| OnlineMCA | - | - | 0.3440 | 0.1323 |
| RQ-VAE | 1.17% | 38.83% | 0.2171 | 0.0689 |
| RQ-Kmeans | 7.03% | 68.08% | 0.2844 | 0.1038 |
| RQ-OPQ | - | **91.91%** | **0.3369** | **0.1194** |

#### A.3.2 RQ-OPQ tokenization Details

Here, we use CUR and ICR as evaluation metrics. The basic codebook size is set to 1024, and the number of codebook layers is set to 3, which aligns with the number of items in the candidate pool. However, e-commerce items have more varied categories and attributes, and RQ-Kmeans tends to prioritize clustering shared prominent features in the former layers. In order to make more concise tokenization, we maintain the capacity of RQ-Kmeans while increasing the codebook size of the former layer to ensure more comprehensive learning of prominent features. As depicted in Table 6, we tested three configurations: (1024,1024,1024), (2048,1024,512), and (4096,1024,256). The codebook size of 4096 achieves higher CUR and ICR, and the Core Keyword Enhancement scheme (\+keywords) shows further improvement. Considering that the search system should encode the entered query similarly and that merchants often increase the number of listed items during global shopping festivals (e.g., 11.11 and 6.18), we further expanded the codebook size to (4096-1024-512). We found that the semantic tokens increased by 11.56% (as $2 \cdot 1.64\%/2.94\% - 1$), and the independent coding rate increased to 43.42% compared to the (4096-1024-512) (\+keywords).

To further improve CUR and ICR, OneRec-V1 (Deng et al., 2025) proposed using full layers balanced k-means. However, for complex fine-grained attributes of items, forcing them into the same cluster in the early stages can lead to hierarchical clustering collapse. As shown in Table 6, the $CUR_{total}$ for the balanced k-means operation on full layers (4096-1024-512$^+$) is much lower than the (\+keywords) configuration. The $CUR$ drastically decreased from 48.95% of $CUR_{L1+L2}$ to 1.64% in $CUR_{total}$, indicating that many similar items were assigned the same ID. Therefore, we propose applying balanced k-means only to the codebook of the third layer to achieve independent encoding of similar items. As shown in (\+l3 balanced), the $CUR_{Total}$ increased from 1.64% to 7.03%, while the $ICR$ improved by 57.15%.

Although RQ-Kmeans can construct hierarchical, learnable SIDs for items, it inevitably discards the residual embedding computed after the last clustering. However, this residual embedding contains the distinctive attributes of each item. Therefore, we further use OPQ for quantizing the unique features. The RQ method handles hierarchical semantics, while PQ is adopted for lateral characteristics. This combined tokenizer can more comprehensively represent the fine-grained features of items, thereby enhancing the relevance constraints for GR models. As shown in Table 7, the two additional SIDs (256-256) generated by OPQ significantly improve the ICR metric and enhance the recall and ranking capabilities of GRs. More detailed testing is introduced in §A.5.

## A.4 IMPLEMENTATION DETAILS

We adopt Bart-B (Lewis et al., 2019) as the base pre-trained model for the testing and online deployment, as it is an efficient model with optimized architectural acceleration, and has been online applied in many scenarios in TEST. Due to commercial confidentiality, we do not disclose the total parameters of the online model here, but it is at least 100 times larger than Bart. The beam search size is set to 512 here to strike a balance between generation quality and latency. The maximum window length is set to n=5. The batch size for SFT and DPO is set to 512 and 128, respectively, with the latter being smaller because the list-wise DPO training takes more samples as inputs. For RQ-OPQ, the number of codebook layers $C = 5$ (3 layers for RQ-Kmeans, and 2 layers for residual OPQ). The codebook size $W$ of each layer is (4096,1024,512—256,256). Some of the hyperparameters will be discussed in the following ablation study. The multi-stage supervised training is conducted every week, RL with the reward system is conducted daily, and the hybrid preference alignment with user interaction data is updated as close to the stream as possible. Actually, RL with a reward system can also be trained every week, as we found it does not bring significant performance gains, except during the global shopping festivals (e.g., 11.11 and 6.18).

## A.5 ABLATION STUDY OF DIFFERENT OPQ TOKENIZATIONS

We examined the impact of different hierarchical quantization encodings on items in Figure 6. As shown in Table 8, we computed two metrics with the top 10 items for quick validation. RQ-OPQ (2/256) is the basic configuration, and RQ-OPQ (4/256) means the residual embedding is tokenized by OPQ (256-256-256-256). RQ-OPQ (4*2/256) means all embeddings (the cluster of three layers and the residual one) are tokenized with OPQ (2/256), then (4*4/256) indicates further quantization. We found that the basic RQ-OPQ (2/256) achieved the highest performance. (4/256) perform weakly with increased sequence length and decoding complexity. The other two configurations were almost entirely ineffective, which is similar to the balanced k-means operation on full layers in § 2.1, as the hierarchical features were not distinctly represented, leading to many items being aggregated under the same SID.

Table 8: Ablation study of different OPQ tokenizations.

| Method | order (30k) | | click (30k) | |
|---|---|---|---|---|
| | HR@10 | MRR@10 | HR@10 | MRR@10 |
| RQ-OPQ (2/256) | 28.42% | 14.15% | 33.69% | 11.94% |
| *-OPQ (4/256) | -2.36% | -1.77% | -2.52% | -1.56% |
| *-OPQ (4*2/256) | -10.20% | -5.57% | -11.77% | -3.84% |
| *-OPQ (4*4/256) | -24.18% | -11.83% | -27.11% | -9.61% |

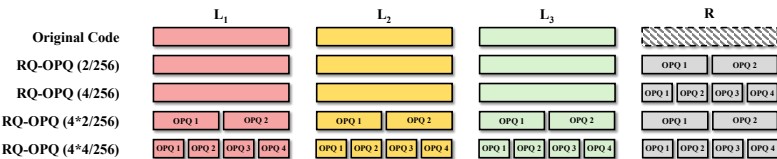

Figure 6: The different hierarchical quantization encodings of items.

## A.6 MANUAL EVALUATION RESULTS FOR ONLINE EXPERIENCE

Last but not least, to ascertain the actual impacts on the online search experience, we conducted additional manual evaluations. We randomly selected 200 queries and extracted 3,200 query-item pairs from identical exposure positions, ensuring all other variables remained constant. We set three metrics as 1) page good rate - an evaluation indicator for the overall user experience, 2) item quality - Check whether the displayed products are counterfeit, have mismatched images and text, or have abnormal prices, and 3) query-item relevance - we engaged experts to rate each pair as "Good" (both subject and core keywords match), "Fair" (only subject matches), or "Bad" (subjects differ). The outcomes of these assessments are presented in Table 9. We can see that OneSearch[2] achieves substantial increases in page good rate by 1.03%, item quality by 2.12%, and query item relevance by 1.87%. The deployment of RQ-OPQ further enhances the relevance of model generation.

Table 9: Manual evaluation results for online experience.

| Metric | Page Good Rate | Item Quality | Q-I Relevance |
|---|---|---|---|
| OneSearch[1] | 0.84% | 1.69% | 1.40% |
| OneSearch[2] | 1.03% | 2.12% | 1.87% |

## A.7 FURTHER STUDIES

**What are the main aspects of the online gains for the OneSearch?** In our analysis, we focused on the dimensions of industry and query popularity. As illustrated in Figure 7, we calculated the CTR relative gains across the top 30 industries. Remarkably, 28 out of 30 industries experienced increases, with an average gain of 2.49%. These results were statistically significant, with P-values below 0.05. Although two industries showed negative effects, these were not statistically significant. Overall, the unified modeling optimization demonstrates substantial potential in addressing the inconsistent objectives of multi-stage processes in MCA systems, benefiting nearly all industries.

As for the query popularity dimension, we divided all prefixes into three categories: top (PV number daily larger than 1,000), middle (larger than 100 and less than 1,000), and long-tail (less than 100). The item CTR relative gains for each were listed in Table 10. Queries of all categories are enhanced with the OneSearch models. These results indicate that the rich semantic and interactive representations induced by keyword-enhanced hierarchical quantization encoding, multi-view behavior sequence, and the preference aware reward system can greatly improve the recognition of e-commerce search for queries of all popularity.

Table 10: Online CTR gains for three query popularity.

| Method | Top | Middle | Long-tail |
|---|---|---|---|
| OneSearch[2] | +1.25% | +2.27% | +1.33% |

**Does OneSearch have stronger reasoning capabilities?** In traditional e-commerce search scenarios, ranking models often involve thousands of features, and the combination of them can obscure some key attributes. Additionally, the structure of common ranking model typically consists of a simple stack of shallow neural networks, resulting in minimal reasoning capabilities. OneSearch, on the other hand, leverages users' long- and short-term sequential information to identify their potential interests and enhances the inference of user search intent through the attention mechanism of transformer structures. For instance, a female user who previously searched for "couple sneakers"

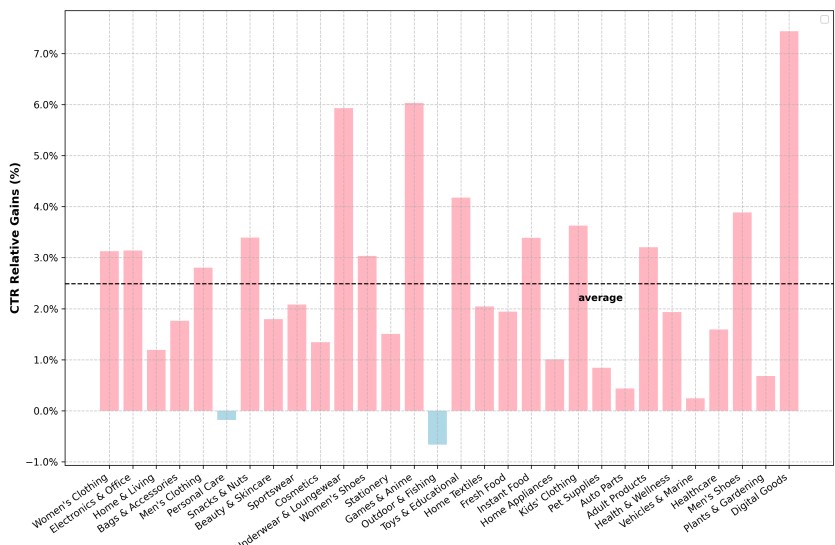

Figure 7: The online CTR relative gains for top 30 industries.

and "Valentine's Day gifts" is likely seeking a pair of rings for both her partner and herself when searching for "silver ring." We observed in real logs that only OneSearch presented the relevant product, which was ultimately purchased by the user.

**How does OneSearch perform for cold-start users and items?** We conducted tests to evaluate the model's performance in cold-start scenarios. Here, we define cold items as those published within the last seven days with no interaction behavior, and cold users as those who have not used the TEST app in the past 90 days. The specific comparison results are demonstrated in Table 11. Compared to the onlineMCA, we found that OneSearch's performance for cold-start items and users has improved by 3.31% and 2.50%, respectively. Both of them are greater than the metrics for warm ones. These results show that OneSearch can handle the cold-start issue well.

Table 11: Online CTR gains for cold-start items and users.

| Object | Warm | Cold | Average |
|--------|--------|--------|---------|
| Item | +2.34% | +3.31% | +2.52% |
| User | +1.11% | +2.50% | +2.41% |

**What potential optimization opportunities will OneSearch explore in the future?** The addition of OPQ-based tokenization can even quickly process new hotwords. We constructed a new keyword offline and added it to the textual descriptions of some items. Without reconstructing a new codebook, OneSearch was still able to generate SIDs for these items during inference. This finding further motivates us to consider online real-time encoding. We will explore in future research, aiming to achieve unified encoding and inference using a single generative model, thereby reducing the gap between scheduled encoding and streaming training phrase. Additionally, aligning user preferences through more robust reinforcement learning and incorporating multi-modal features (such as images and videos) for items can further enhance the reasoning capabilities.

