# OpenReview forum: "OneSearch: A Preliminary Exploration of the Unified End-to-End Generative Framework for E-commerce Search"
_ICLR.cc/2026/Conference — ICLR 2026 Conference Withdrawn Submission_

### Official Review · Reviewer_kUbD · 2025-10-24

**Soundness:** 3
**Presentation:** 3
**Contribution:** 3
**Rating:** 4
**Confidence:** 4

**Summary:**

The paper introduces OneSearch, a novel end-to-end generative framework for e-commerce search that replaces traditional multi-stage cascading architectures (MCA) with a unified model. The core idea is to generate item ranking list directly from user-query information, thus addressing the limitations of fragmented computation and optimization conflicts in MCA systems.

The proposed framework integrates three key innovations:
1. Keyword-enhanced Hierarchical Quantization Encoding: A hybrid RQ-OPQ tokenizer that preserves both shared and distinctive item features to improve semantic representation.
2. Multi-view User Behavior Sequence Injection: A method to encode user preferences using short- and long-term behavior signals, enhancing personalization.
3. Preference-Aware Reward System: A multi-stage fine-tuning process with an adaptive reward model to align generated rankings with user preferences.

Extensive offline evaluations and online A/B testing demonstrate significant improvements on multiple metrics.

**Strengths:**

1. The paper presents the first industrial-deployed end-to-end generative framework for e-commerce search, moving beyond the limitations of multi-stage architectures. This represents a significant contribution to both research and practice.
2. The paper provides comprehensive offline and online evaluations. Improvements in online metrics are significant and the proposed method claims to reduce the OPEX by a large extent.

**Weaknesses:**

1. The paper writing is a little bit hard to follow,  especially the tokenization part. The RQ-OPQ part does not give very clear description, the content in appendix A.3 includes a lot of details on experiment/implementation, but does not describe the method itself. And OneSearch includes many components, as shown in Figure 2, it is hard to get the major point of the proposed method.
2. No experiment results on open-source datasets. As the REPRODUCIBILITY STATEMENT pledges that the codebase will be released, perhaps it's better to provide results on open datasets.
3. Training details are not shown in the paper, like training cost, online update method, etc.

**Questions:**

1. The RM seems like a traditional ranking module, which seems the most critical part to the online AB test improvements shown in Table 4. If OneSearch still needs a "three-tower SIM" as the reward model to rerank the list, it is not fully "end-to-end". The proposed model is actually recall+pre-ranking?
2. Why the results in Table 2 uses HR@350 for both online MCA and w/o ranking? 350 seems like a length for testing pre-ranking or recall, not very suitable for the ranking model. Considering the proposed OneSearch (w/ RM reranking) is essentially recall+pre-ranking, its performance on HR@350 is not as good as "w/o ranking".
3. In section 3.4, the paper claims to reduce the OPEX for 24.6%. How is it calculated? Because OneSearch with RM reranking still use a traditional ranking model with the recall&pre-ranking part replaced by Transformer structure, it is still a multi-stage complex system.

---

> ### Author Response · Authors · 2025-11-17
> **Response to Weaknesses:**
>
> Thank you very much for reading carefully. We will provide detailed responses to your proposed weaknesses and questions separately.
> 1. **The RQ-OPQ description**. Due to paper length requirements, the RQ-OPQ method is detailed in Appendix A.3 and A4. In short, we found the common SID tokenizers, such as RQ-VAE, VQ-VAE, and RQ K-means, focus on encoding shared features among similar items, which can result in the loss of distinctive features for each item. Therefore, we proposed using OPQ to tokenize the residual embedding after the last RQ layers. The former RQ (4096,1024, 256) handles commonly hierarchical semantics, and OPQ (256, 256) is used for lateral distinct characteristics. The strategy improved the ICR metric and enhanced the recall and ranking capabilities of GR significantly. Furthermore, the RQ-OPQ tokenization has not only achieved remarkable results in end-to-end generative algorithms, but also been applied in other recent research works [1,2], verifying its effectiveness in tasks such as the similar item recall, precise-ranking. We apologize again for the difficult-to-understand descriptions in the paper, and promise to simplify them in the revised version. **The main contribution of OneSearch**. Generative retrieval (GR) has been extensively studied in recommendation systems, such as Tiger[3] and OneRec[4], but there is still no industrially deployable solution in search systems. This is because search has strong relevance constraints caused by queries, meaning that all items returned by the system must satisfy the intent of the query. In addition, the system must be able to accurately model user preferences to achieve personalized item ranking. These issues require that a GR model that can be deployed must first do a good job of efficiently representing items, ensuring that the encoding can both hierarchically describe the commonalities of items and model their unique fine-grained features; at the same time, it must be able to accurately understand user preferences based on historical behavior sequences in order to do a good job of items ranking and bring about business conversion. The dual requirements of semantics and conversion are not addressed by previous GR models. OneSearch addresses this via: 1) Keyword-enhanced hierarchical quantization for item tokenization, 2) Multi-view user behavior injection for preference modeling, and 3) Preference-aware rewards for personalized ranking. All these contribute to the first industrial-deployed end2end generative framework for e-commerce search, with more revenue and less expenditure.
> 2. **No experiment results on open-source datasets**. Currently, there is no publicly available e-commerce search dataset in the community that is constructed from real user behavior logs and includes user profiles, search queries, historical behavior, and interactions with each product, because this involves user privacy and business secrets. Common e-commerce works are all validated on their own platform's closed datasets, such as R4R [5], TB-HNS [6], and C2T-ID [7], etc., which do not publicly disclose their own datasets. This is fundamentally different from general retrieval tasks such as NQ [8] and MS-MARCO [9], and rare shopping queries dataset, like Amazon ESCI [10], focusing on the semantic matching, which do not involve complex user historical behavior and distinct business conversion levels. We plan to open-source our evaluation dataset and core codes under the company's limitation, in order to fulfill ICLR's REPRODUCIBILITY STATEMENT pledges and provide a public benchmark for further exploration in academia.
> 3. **No Training details and online update method**. The training details are described in Appendix A.4 Implementation details, the SFT loss is the basic next-token-prediction log-likelihood (NLL) loss, and the optimization objective for each stage is shown in Sec 2.4 Preference aware reward system part. The online update method is detailed in Figure 2. We apologize again for the chaotic descriptions of these training configurations, and promise to organize them in the revised version.
>
> Brief references with character limitations:
>
> [1] SMILE: SeMantic Ids Enhanced CoLd Item Representation for Click-through Rate Prediction in E-commerce SEarch.
>
> [2] OneVision: An End-to-End Generative Framework for Multi-view E-commerce Vision Search.
>
> [3] Recommender systems with generative retrieval.
>
> [4] OneRec Technical Report.
>
> [5] Retrieval-in-the-Chain: Bootstrapping Large Language Models for Generative Retrieval.
>
> [6] Taxonomy-based Negative Sampling In Personalized Semantic Search for E-commerce.
>
> [7] C2T-ID: Converting Semantic Codebooks to Textual Document Identifiers for Generative Search.
>
> [8] Natural questions: a benchmark for question answering research.
>
> [9] Ms marco: A human generated machine reading comprehension dataset.
>
> [10] Shopping queries dataset: A large-scale ESCI benchmark for improving product search.

---

> ### Author Response · Authors · 2025-11-17
> **Response to Questions:**
>
> First of all, we claim that OneSearch is **the end-to-end generative model**, as it was deployed in the mall search platform with the replacement of the entire "recall+pre-ranking+ranking" stages, other than the only "recall+pre-ranking" modules.  Sec 3.3 shows three online A/B results of OnlineMCA (w/o ranking stage),  OneSearch (w/o RM), and OneSearch (w/ RM). They are listed in order to verify the importance of the ranking stage in OnlineMCA, and that our OneSearch (w/o RM) has inherent ranking capabilities, achieving the comparable performance to the entire OnlineMCA (recall+pre-ranking+ranking). The results of OneSearch (w/ RM) are shown mainly to further validate the SIM-based RM module's ability in personalized modeling and preference ranking. The real deployed version is the pure generative model without any other treatment.
> Specifically, the total adaptive reward system contains two-phase alignment (details in Sec 2.4): 1) **SIM-based Reward Model**. It differs from the OnlineMCA ranking with  **much less feature dimension** (only takes user ID, query, user behavior sequence, and user profile as input), but the Online Ranking model has thousands of features, as well as **more sample strategy**, it also takes the items of the same category clicked in recommendation scenarios as training samples. This model is updated only once every three days, which is very different from ranking models that require streaming updates, as it is designed to instruct the initial OneSearch model to adhere to the online distribution and learns ranking capabilities of the MCA. The second phase, 2) **User interaction-based Reward learning**, needs no additional parameters. It used the online real data to update OneSearch self as close to streaming as possible. This design aims to overcome online limitations and further leverage the inference capabilities of generative models. So the OneSearch (w/ RM) cannot be directly used online as it inherently limits OneSearch’s ability to exceed Online MCA’s performance ceiling with untimely training.
>
> Then, we provide detailed responses to your proposed questions.
> 1. **RM seems like a traditional ranking module**. The SIM-based RM is not a subsequent ranking after pure OneSearch model. The SIM is trained per 3 days for instructing the initial OneSearch learning the basic ranking capabilities of the MCA. In practice, this model can be trained only once and then never used again. We have tested the SIM-trained-once version in the online system (OnSearch updated over one month). It can get much better performance than OnlineMCA as time goes by:
>
>
> |          | Item CTR | PV CTR | PC CVR | Buyer  | Order |
> |----------|----------|--------|--------|--------|-------|
> | first week  | +1.67%*  | +1.43% | +1.98% | +1.91%* | +1.10%* |
> | second week | +1.17%*  | +0.58%* | +3.84% | +1.98% | +2.69% |
> | third week  | +1.81%  | +1.46%* | +3.52% | +2.49% | +3.17% |
> | forth week  | +3.30%  | +1.15% | +4.43% | +2.67% | +3.33% |
>
> Above is the revenue gains of pure OneSearch compared to OnlineMCA (09.15-10.13). The digits* mean that the statistical significance (P-value) is larger than 0.05, as the data are not yet significant.
>
> 2. **N=350** for the metrics Recall@N and MRR @N in all testings was intentionally set. As 350 is the 99th-percentile viewed item number of one search PV in our online system (99% of users' search PV will reach). This means that, excluding the very few unstable requests caused by machine malfunctions or network latency, the actual experience for almost all users is. This is a monitoring indicator for online system stability.
> 3. **OPEX calculation**. As the real deployed version is the pure GR model, and the RM is only used for training, the reduction of OPEX for 24.6% is only for the OneSearch model, compared to the multi-stage OnlineMCA.
>
> Thank you again for your careful review. If you have any further questions, please feel free to ask, and we will provide you with a thorough and detailed answer.

---

> > ### Comment · Reviewer_kUbD · 2025-11-26
> >
> > Thank you for the detailed response. It addresses some of the concerns.
> > There are some points that are still confusing.
> > 1. As OneSearch has inherent ranking capabilities and using SIM as a reranking model will boost the performance further, is the baseline OnlineMCA also has a re-ranking stage?
> > 2. In Table4, we could find that OneSearch w/o RM reranking has basically the same online performance with MCA. Appart from the online metrics, is there other indicators such as inference overhead or scaling evidence that could prove the dense Transformer architecture is better than the multi-stage systems? The current Section3.4 results seems a little bit vague. MFU is important but it does not have direct influence on user experience or system cost. Because the MFU increase simply comes from dense Transformer require more compute and the original sparse ID system does not.
> > 3. In Table2，why "w/o ranking" has better HR@350 than OnlineMCA? Is the HR here actually recall rate?

---

> ### Author Response · Authors · 2025-11-26
> **Further Response**
>
> Thank you for your careful review. We will provide detailed responses to your proposed points.
>
> 1. **Dose baseline OnelineMCA has a re-ranking stage?** The ranking stage of OnlineMCA contains the basic point-wise ranking (ctr\cvr) and furhter list-wise reranking sub-stage. We reported in the paper that OneSearch (with RM module) can boost the performance of pure OnlineMCA. Furthermore, we also reported above that the pure OneSearch (w/o RM module) online can get the much better conversion that the streaming (real-time) training. Here we repeat the A/B testing results above,  of which the base is the streaming OnlineMCA (with point-wise ranking and further list-wise reranking), and the testing is **the pure OneSearch (w/o RM module)**. Specifically, the pure OneSearch took the query, and user information as input, and outputed the item_id directly, without any further process.
>
> |          | Item CTR | PV CTR | PC CVR | Buyer  | Order |
> |----------|----------|--------|--------|--------|-------|
> | first week  | +1.67%*  | +1.43% | +1.98% | +1.91%* | +1.10%* |
> | second week | +1.17%*  | +0.58%* | +3.84% | +1.98% | +2.69% |
> | third week  | +1.81%  | +1.46%* | +3.52% | +2.49% | +3.17% |
> | forth week  | +3.30%  | +1.15% | +4.43% | +2.67% | +3.33% |
>
> 2. **other indicators**  In Sec 3.4, we actually provided two metrics to demonstrate the superiority of the dense transformer architecture. 1) The first is **MFU (Model FLOPs Utilization)**, which tests the machine's GPU utilization efficiency. A higher metric indicates better model-machine utilization and fewer machines required for the same consumption. Considered that the waste of MFU at each stage of Online MCA, the end-to-end OneSearch can significantly improve this key metric, as OneSearch achieves a 27.32% MFU improvement compared to onlineMCA's 3.26%, an 8.4× improvement. 2) A significant contribution by the improved MFU is the substantial reduction in computational resources required to handle the same search PV (Page Views), which is our second testing metric, **OPEX (operational expenditure)**. The reduction of 75.40% of OPEX specifically means that the computational cost of OnlineMCA (including data storage, CPU, and GPU for recall, pre-ranking, and ranking stage) is E1, while the computational cost of OneSearch to complete the same daily search PV is E2. E2/E1 = 24.6%, indicates a significant reduction in both storage and computational costs, and higher business efficiency.
>
> Another metric that effectively evaluates the efficiency is inference time. The complete inference time of one search PV for OneSearch is only one-twentieth that of OnlineMCA. Specifically, OneSearch can complete almost all inference in less tens of milliseconds, a significant optimization compared to the latency required by OnlineMCA. The online version OnlineMCA is based on the Bart model. Further testings on the decode-only model is also conduct on the scaling Qwen3-0.6B and 1.7B, we also found the higher MFU, lower OPEX and inference latency compared to OnlineMCA. We will further highlight these results and improve the table layout in the latter version.
>
> 3. **higher HR@350** All the HR@350, and MRR@350 are the actual values for online search pv. The method "w/o ranking" means the testing version of OnlineMCA only have "recall and pre-ranking" stage. Compared to full version of OnlineMCA with additional "ranking" stage, the higher HR@350 and lower MRR350 is accurate. As the final "ranking" stage focus on improving the model’s personalized ranking capability with thousands of interactive features, some potential items with low predicted CTR (Click-Through-Rate) and CVR (Conversion Rate) would be ranked later, which is a common but bad phenomenon in multi-stage architectures.

---

> ### Author Response · Authors · 2025-11-27
> **Further Rebuttal**
>
> Thank you again for your careful review. If you have any further questions, please feel free to ask, and we will provide you with a thorough and detailed answer.

---

### Official Review · Reviewer_XXwD · 2025-10-25

**Soundness:** 2
**Presentation:** 1
**Contribution:** 2
**Rating:** 4
**Confidence:** 3

**Summary:**

This paper introduces OneSearch, an end-to-end generative framework designed to replace traditional multi-stage architecture in e-commerce search. The authors propose a unified model that handles recall, pre-ranking and ranking. The core contributions seem to include a keyword-enhanced hierarchical quantization scheme, a novel strategy to input user behavior sequences, and a preference-aware reward alignment system. The authors report significant improvements in online A/B tests on a large-scale e-commerce platform, while achieving better GPU utilization and lower operating cost.

**Strengths:**

- The general idea of the paper is to unify e-commerce search into a single model, which is a highly interesting and relevant research question.
- The proposed OneSearch architecture seems to be highly optimized and is comprised of several meticulously crafted and fine-tuned parts, which is an impressive engineering accomplishment
- The model achieves impressive results over the previous production model, and several ablations validate individual aspects of it.
- In particular, the  proposed RQ-OPQ tokenization method appears to be a novel and effective idea to capture relevant remaining information after quantization.

**Weaknesses:**

While the paper tackles an important problem and presents promising results, it is difficult to follow and lacks clarity in key parts. It additionally reads like a system report rather than a research paper a lot of the time, making it difficult to assess its scientific novelty and contribution. In particular,

- The paper's clarity of writing could be greatly improved. In parts, the text is dense and convoluted, making it difficult to follow. There are several abbreviations that are used without proper introduction, forcing the reader to look up their definitions elsewhere and preventing the paper from being a stand-alone text. Since the paper proposes a pretty complex and intricate architecture, these shortcomings impact its readability and, to some extent, obscure its actual contributions.
- The proposed OneSearch method consists of a lot of components, and the paper does not always clarify which of these are novel contributions, and which are taken from previous work. These omissions cause the method section to partially read like a system report rather than a research paper.
- Some key methodological details are missing. For example, the loss functions in Section 2.1 are not self-contained. Terms like the "hard sample relevance correction loss" are introduced without any definition. While many of the non-explained terms are standard in retrieval systems, these omissions and the resulting ambiguity make it difficult to fully understand and appreciate the proposed method.
- The experimental evaluation lacks transparency. The dataset is described only as "TEST," which is not informative. The baseline, "onlineMCA," is a proprietary system, making it impossible to judge whether the reported gains are due to the novelty of OneSearch or simply differences in engineering and feature tuning. The paper sometimes states that OneSearch improves metrics by a certain percentage without clearly stating what it's being compared against in that specific context.
- Figures 2 and 3 are very large and cause physical lag on some PDF readers. Additionally, Figure 2 contains a lot of information, resulting in a somewhat overwhelming chart that does little to clarify the model.

**Questions:**

- Can the authors clarify the core scientific contributions of this paper, and differentiate it from the (impressive) engineering effort of the proposed method? What exactly are the main advances compared to existing work in dense and generative retrieval? What are durable takeaways for the wider research community?
- Related to this, how does the paper compare to recent generative frameworks like OneRec, which also unifies the recommendation pipeline, and GRAM, which applies generative retrieval to e-commerce. Could the authors elaborate on the key differences, and the reasons why OneSearch is particularly suited to the e-commerce setting?
- A significant part of the methodology appears to be highly tuned to your platform's data and infrastructure (e.g., the 18 structured attributes from NER, the specific six-level user behavior hierarchy). Which of the design choices of the method are specific to the chosen platform, and which would be expected to translate to other settings?
- The experiment section is based on the comparison to and relative improvement over the "onlineMCA" baseline. While I understand that the specifics are proprietary, could the authors provide a higher-level summary of the baseline? Similarly, could the method be, in part, compared to other existing work?

---

> ### Author Response · Authors · 2025-11-19
> **Response to Weaknesses:**
>
> 1. **Clarity of Writing and Abbreviations**. We apologize for the lack of clarity in key sections. In the revised manuscript:
>     - Some critical abbreviations (e.g., CTR = Click Through Rate, CVR = ConVersion Rate, PV = Page View) are explicitly defined upon first use.
>     - Complex sections (e.g., architecture descriptions in Sec. 2) are restructured with simplified language to enhance readability and conciseness.
> 2. **Proposal Novelty or Previous Work**. We will explicitly demarcate novel contributions from previous work. This is detailed in the response to Question 1.
>
> 3. **Missing Methodological Details**. The "hard sample relevance correction loss" (used in Sec. 2.1) is a typical MSE loss. Which mainly addresses the hard and easy-misunderstanding pairs. For pairs with a threshold similarity, we use LLM to score the relevance based on full context information of query and item, and then let the distilled BGE model fit this score.
> 4. **Experimental Transparency**.
>     - Datasets: Currently, there is no publicly available e-commerce search dataset in the community that is constructed from real user behavior logs and includes user profiles, search queries, historical behavior, and interactions with each product, because this involves user privacy and business secrets. Common e-commerce works are all validated on their own platform's closed datasets, such as R4R [1], TB-HNS [2], and C2T-ID [3], etc., which do not publicly disclose their own datasets. However, we plan to open-source our evaluation dataset and core codes under the company's limitation, to fulfill ICLR's REPRODUCIBILITY STATEMENT pledges and provide a public benchmark for further exploration in academia.
>     - Improvement: Tables 3, 4, and 7, respectively, verify the performance improvements of the Mu-Seq, PARS, and KHQE modules. The overall improvement is shown in the last row of Table 4.
>     - Metrics: All reported gains are relative to onlineMCA unless stated otherwise.
> 5. **Figure Optimization**. Thanks for your advice. We simplified Figures 2 and 3 by resizing and removing redundant labels.
>
> **Brief references with character limitations**:
>
> [1] Retrieval-in-the-Chain: Bootstrapping Large Language Models for Generative Retrieval.
>
> [2] Taxonomy-based Negative Sampling In Personalized Semantic Search for E-commerce.
>
> [3] C2T-ID: Converting Semantic Codebooks to Textual Document Identifiers for Generative Search.

---

> ### Author Response · Authors · 2025-11-19
> **Response to Questions:**
>
> 1. **Core Scientific Contributions**.
>     - **Search vs. Recommendation complexity**: Unlike recommendation systems, where both inputs and outputs are closed-vocab items (Tiger, OneRec), e-commerce search faces open-vocab queries with strict relevance constraints, i.e., all returned items are strongly correlated with the query and fully meet the user's personalized preferences. This requires the generative model to have hierarchical and distinct tokenization for each item and to have accurate modeling of user behavior sequences.
>     - **Novel contributions addressing search-specific challenges**: a) **KHQE with RQ-OPQ**: Our approach specifically addresses the dual requirements of **hierarchical semantic encoding** and **distinctive fine-grained features preservation**, which is critical for query-item relevance in search. The 91.91% ICR (vs. 68.08% for RQ-Kmeans alone, Table 7) demonstrates it is not a simple adaptation. b) **Keyword enhancement**: This tackles the **noisy item information** problem unique to e-commerce search, which neither Tiger nor OneRec addresses. c) **Multi-view behavior injection**:  Our explicit/implicit sequence design specifically addresses the challenge of **inferring latent search intent from concise queries** (typically 2-3 keywords), which differs fundamentally from session-based recommendation.
>     - **First industrial deployment for search**: In Sec. 4 and A.2, existing GR works (GenR-PO, GRAM) only optimize **partial stages**. OneSearch is the **first to unify all three stages (recall + pre-ranking + ranking)** in e-commerce search.
> 2. **Comparison to OneRec and GRAM.**
>     - **vs. OneRec**: Figure 5 illustrates the essential difference. OneRec unifies recommendation stages to provide diverse items to the user without a query-item relevance constraint, while OneSearch is strictly limited by this constraint. A traditional search system not only requires a cascading architecture but also an additional relevance module to specifically address this constraint. Therefore, OneSearch not only unifies the cascaded framework but also eliminates the need for relevance models. OneSearch addresses this via KHQE and query-item co-occurrence in multi-stage SFT, crucial for search where relevance > personalization.
>     - **vs. GRAM**: GRAM is served for search too. The difference is that GRAM can only replace the recall and pre-ranking processes, but not the ranking process, which is extremely sensitive to user preferences. However, OneSearch can unify the entire cascading process via Mu-Seq and PARS that significantly improve the model's personalized ranking capabilities.
> 3. **Platform-Specific vs. Generalizable Components**. Item data in e-commerce is highly structured, and the 18 attributes are also the fundamental components for other e-commerce platforms. Furthermore, the six-level user behavior hierarchy with common signals is reusable in e-commerce scenarios. After removing the purchase signal, it can be further promoted to a generic content platform. KHQE’s quantization, Multi-view sequences, and PARS’s tiered rewards are universal for other end-to-end scenarios or separated stages [1,2].
> 4. **Baseline Clarification and Extended Comparisons**.
>     - **Baseline Clarification**: OnlineMCA represents the traditional Multi-stage Cascading Architecture (in Figure 1.b) in recommend and search systems, which progressively filters items through recall (10^9 items), pre-ranking (10^4 items), and ranking (10^2 items) stages. The recall phase typically employs index-based and embedding-based methods. Although both pre-ranking and ranking are sorting modules, they mainly adopt the dual-tower and SIM models, respectively, due to the different quantities of items they handle. Both are constrained by corresponding relevance models to ensure that items ranked higher satisfy the query's requirements. More importantly, each module of OnlineMCA is not a single model, but rather a combination of many SOTA models, to achieve good results in handling different cases.
>     - **Extended comparisons**: We cannot compare it with existing works like GRAM, for it is closed. However, we provide detailed offline and online ablation results. In Table 4, the MCA without ranking experiment demonstrates that removing ranking results in a significant performance drop, while $OneSearch_{RM}^2$ not only withstood this huge drop but also brought additional improvements. In Table 2, we found that OneSearch's gains come from stronger ranking capabilities than pre-ranking and a higher recall than ranking, indicating more robust synthesized search performance.
>
> **Brief references:**
>
> [1] SMILE: SeMantic Ids Enhanced CoLd Item Representation for Click-through Rate Prediction in E-commerce SEarch.
>
> [2] OneVision: An End-to-End Generative Framework for Multi-view E-commerce Vision Search.
>
> Thanks again for your careful review. If you have any further questions, please feel free to ask, and we will provide detailed answers.

---

> > ### Comment · Reviewer_XXwD · 2025-11-26
> > **Response to Authors**
> >
> > I thank the authors for their detailed discussion and for incorporating several of my suggestions into the paper. From my point of view, the provided version seems easier and more enjoyable to read, and several of my questions and concerns have been clarified. I still think that the figures are a bit too dense and contain more details than probably necessary, but can see that having a "complete" figure of the architecture is a valid design choice.
> >
> > I especially appreciate the explicit mention of core contributions, as this helps to meet the paper on its own terms and distinguish it from related work. I think that the provided response paints a clearer picture of the contributions than the current paper does, and would carefully suggest to add some of the points made to the revised paper (if there isn't a reason against it). However, as is, most of my questions and concerns have been clarified, and I am happy to raise my score accordingly.

---

### Official Review · Reviewer_ofXQ · 2025-10-27

**Soundness:** 3
**Presentation:** 2
**Contribution:** 2
**Rating:** 4
**Confidence:** 2

**Summary:**

This paper presents OneSearch, an end-to-end generative framework for e-commerce search that aims to replace traditional multi-stage cascading architectures (MCA). The framework introduces (1) Keyword-enhanced Hierarchical Quantization Encoding (KHQE) that preserves hierarchical semantics while maintaining query-item relevance, (2) a multi-view user behavior sequence injection strategy incorporating both short-term and long-term user preferences, and (3) a Preference-Aware Reward System (PARS) with multi-stage supervised fine-tuning and adaptive reward modeling. Experimental results suggest that OneSearch achieves significant improvements in online A/B tests while reducing operational costs.

**Strengths:**

- The paper demonstrates substantial practical value with comprehensive online A/B testing results showing statistically significant improvements across multiple metrics. The successful deployment across multiple search scenarios is also meaningful.

- The KHQE module with RQ-OPQ tokenization and keyword enhancement addresses specific challenges in e-commerce search, such as noisy item information and strict relevance constraints. The multi-view behavior sequence injection strategy effectively handles both explicit short-term and implicit long-term user preferences, which is crucial for personalized search.

- The paper provides thorough offline evaluations, detailed ablation studies, and extensive online testing. The analysis covers various aspects like cold-start scenarios, different query popularities, and manual evaluation results.

**Weaknesses:**

- While the overall framework is novel, individual components largely build upon existing techniques. The core generative retrieval paradigm follows and adapts established patterns from Tiger and OneRec.

- The paper lacks discussion of scenarios where OneSearch underperforms or fails. There's limited analysis of the trade-offs between the unified approach and specialized multi-stage systems, particularly for edge cases or specific query types. The paper also doesn't adequately address potential scalability concerns or discuss how the approach might degrade with extremely large item catalogs.

**Questions:**

please refer to the weakness section

---

> ### Author Response · Authors · 2025-11-19
> **Response to Questions:**
>
> Thank you for your constructive feedback. We provide detailed responses to address your concerns:
>
> **W1: Novelty of individual components**
>
> We respectfully clarify that while OneSearch builds upon the generative retrieval paradigm, the key distinction lies in **adapting these techniques to the fundamentally different challenges of e-commerce search** versus recommendation. As we emphasized in Appendix A.2:
> 1. **Search vs. Recommendation complexity**: Unlike recommendation systems where both inputs and outputs are closed-vocabulary items (Tiger, OneRec), e-commerce search faces **open-vocabulary** queries with strict relevance constraints. That is, we must ensure that all returned items are strongly correlated with the query and fully meet the user's personalized preferences. This requires the generative model to have hierarchical and distinct tokenization for each item and to have accurate modeling of user behavior sequences.
> 2. **Novel contributions addressing search-specific challenges**:
> 	○ **KHQE with RQ-OPQ**: Unlike pure semantic ID tokenization in OneRec, our approach specifically addresses the dual requirements of **hierarchical semantic encoding** and **distinctive fine-grained features preservation**, which is critical for query-item relevance in search. The 91.91% ICR (vs. 68.08% for RQ-Kmeans alone, Table 7) demonstrates this is not a simple adaptation.
> 	○ **Keyword enhancement**: This directly tackles the **noisy item information** problem unique to e-commerce search. Neither Tiger nor OneRec address this challenge.
> 	○ **Multi-view behavior injection**: While user modeling exists in recommendation, our explicit/implicit sequence design specifically addresses the challenge of **inferring latent search intent from concise queries** (typically 2-3 keywords), which differs fundamentally from session-based recommendation.
> 3. **First industrial deployment for search**: As stated in our Related Works (Section 4 and Appendix A.2), existing GR works in search (GenR-PO, GRAM) only optimize partial stages (recall, pre-ranking). OneSearch is **the first to unify all three stages (recall + pre-ranking + ranking)** in e-commerce search, which has fundamentally different challenges than recommendation.
>
> **W2: Failure scenarios and trade-off analysis**
>
> This is a valuable point. As we discussed detailed in Sec A.7. We found that 28 out of top 30 industries could get the experienced increases of CTR, with an average gain of 2.49%. Also, queries of top, middle, and long-tail are all enhanced as OneSearch can greatly improve the recognition of e-commerce search for queries of all popularity. For instance, a female user who previously searched for ”couple sneakers” and ”Valentine’s Day gifts” is likely seeking a pair of rings for both her partner and herself when searching for ”silver ring.” We observed in real logs that only OneSearch presented the relevant product, which was ultimately purchased by the user. Furthermore, we also showed that OneSearch performed better for cold-start users and items in the last testings.
>
> As we found that, OneSearch can make a better understanding and item recommendation for Clear intent query search, that OnlineMCA on almost all scenes. However, we also found that **OneSearch and MCA perform poorly with complex queries lacking explicit intent**. For example, queries like "Monday OOTD" and "makeup recommendations for beginners" don't clearly specify a particular item category. Instead, they require complex intent decomposition,  action execution and combined recommendations based on the user's historical behavior,  which may necessitate inference models with intent decomposition and task planning capabilities, or multi-agent system. **This is beyond the capabilities of current end-to-end generative models** and is a key area of ​​our future research.
>
> Thank you for raising such a valuable point. We will discuss this issue in the revised version.

---

### Official Review · Reviewer_5bLT · 2025-11-03

**Soundness:** 3
**Presentation:** 3
**Contribution:** 4
**Rating:** 6
**Confidence:** 2

**Summary:**

This work introduces a generative search framework with four key contributions:
1. Keyword-Enhanced Hierarchical Quantization Encoding: an encoding method that balances contextual understanding and collaborative information, while reinforcing relevance constraints
2. Multi-View Behavior Sequence Injection: a strategy that integrates user behavior patterns into ID representations, using both explicit and implicit prompts to improve the model’s reasoning about user preferences.
3. Preference-Aware Reward System: a personalized ranking mechanism utilizing multi-stage supervised fine-tuning (SFT) and adaptive reward modeling to enhance recommendation precision.
4. OneSearch end-to-end unified framework: deployed the first industrial-level, end-to-end generative search system that unifies all the above techniques for an eCommerce search setting with a compelling online A/B test results.

There are some details left out in the paper:
1. it would be valuable to discuss what challenges for production rollout, especially from infra perspective.
2. Some experiments shared, it would be better to explain more rationals/intuitions for them. For example, in Table 2, OnlineMCA w/o ranking seems to have higher value for HR@750 for both order (30k) and click(30k) datasets, But the bold results are RQ-OPQ+Adaptive RS, inferring that we need to balance precision with recall. This was not very clear so maybe providing explanation to make it more clear. In Table 4, it seems that the combine effect of full optimizations +  RM is smaller than each individual ones.
3. It was not clear on why online MCA was chosen as control for online A/B test, and if other benchmarks considered and why not.

**Strengths:**

This work introduces a generative search framework with four key contributions:
1. Keyword-Enhanced Hierarchical Quantization Encoding: an encoding method that balances contextual understanding and collaborative information, while reinforcing relevance constraints
2. Multi-View Behavior Sequence Injection: a strategy that integrates user behavior patterns into ID representations, using both explicit and implicit prompts to improve the model’s reasoning about user preferences.
3. Preference-Aware Reward System: a personalized ranking mechanism utilizing multi-stage supervised fine-tuning (SFT) and adaptive reward modeling to enhance recommendation precision.
4. OneSearch end-to-end unified framework: deployed the first industrial-level, end-to-end generative search system that unifies all the above techniques for an eCommerce search setting with a compelling online A/B test results.

**Weaknesses:**

Overall it was a good paper. There are some details left out in the paper:
1. it would be valuable to discuss what challenges for production rollout, especially from infra perspective.
2. Some experiments shared, it would be better to explain more rationals/intuitions for them. For example, in Table 2, OnlineMCA w/o ranking seems to have higher value for HR@750 for both order (30k) and click(30k) datasets, But the bold results are RQ-OPQ+Adaptive RS, inferring that we need to balance precision with recall. This was not very clear so maybe providing explanation to make it more clear. In Table 4, it seems that the combine effect of full optimizations +  RM is smaller than each individual ones.
3. It was not clear on why online MCA was chosen as control for online A/B test, and if other benchmarks considered and why not.

**Questions:**

Overall it was a good paper. There are some details left out in the paper:
1. it would be valuable to discuss what challenges for production rollout, especially from infra perspective.
2. Some experiments shared, it would be better to explain more rationals/intuitions for them. For example, in Table 2, OnlineMCA w/o ranking seems to have higher value for HR@750 for both order (30k) and click(30k) datasets, But the bold results are RQ-OPQ+Adaptive RS, inferring that we need to balance precision with recall. This was not very clear so maybe providing explanation to make it more clear. In Table 4, it seems that the combine effect of full optimizations +  RM is smaller than each individual ones.
3. It was not clear on why online MCA was chosen as control for online A/B test, and if other benchmarks considered and why not.

---

> ### Author Response · Authors · 2025-11-19
>
> Thank you very much for reading carefully. We will provide detailed responses to your proposed weaknesses.
>
> 1. **The challenges faced in promotion**. The primary challenge for deploying OneSearch in production lies in managing the trade-off between model capacity and online resource consumption. While we indeed observe a clear scaling-law pattern in the search domain—where larger models consistently deliver superior performance—such gains come at the cost of significantly increased online serving requirements, including GPU/CPU utilization, and network bandwidth. Although OneSearch reduces overall operational expenditure (OPEX) to 24.60% of that of the existing multi-stage pipeline (recall, pre-ranking, and ranking), the balance between model size, latency, and serving cost remains a central concern.To mitigate this issue, our future efforts will focus on enhancing the model’s reasoning capabilities (e.g., explicit and implicit reasoning) to further improve effectiveness without increasing parameter size. We believe this direction is essential for achieving a more scalable, resource-efficient, and practically deployable end-to-end search system.
>
> 2. **Explaination of our experiments**. We sincerely apologize that, due to paper limitations, a detailed explanation of the experimental results was not provided. As for Table 2, pre-ranking module is designed to ensure that as many high-quality candidates as possible enter ranking stage, so, OnlineMCA w/o ranking achieve high HR@350 but a low MRR@350. Conversely, the ranking model aims to accurately estimate the candidate's CTR, therefore, it often stacks a large number of complex model structures (DIN, DSSM, MMoE, DCN), resulting in a high MRR@350 and a low HR@350. OneSearch essentially achieves a balance between pre-ranking and fine ranking.
>
> Regarding the results in Table 4, I think there is a latent misunderstanding. In fact, the combined setting OneSearch2(full optimizations) + RM achieves the best performance among all variants. The reviewer may have accidentally referred to the row of OneSearch2 w/o RM, which corresponds to the full optimizations only setting. To prevent such confusion, we will further highlight this result and improve the table layout in the revised version.
>
> 3. **No experiment results on open-source datasets**. We would like to clarify that OneSearch is the first unified end-to-end model deployed in online industrial search system. Unlike existing models, which are designed for specific stages of the traditional multi-stage architecture (e.g., recall, pre-ranking, or ranking), OneSearch replaces the entire pipeline with a single end-to-end framework. Therefore, these stage-specific models are not directly comparable baselines for evaluating a unified architecture. **More importantly, each module of OnlineMCA is not a single model, but rather a combination of many SOTA models, with the aim of achieving good results in handling different cases. For the two reasons mentioned above**, our online A/B test uses the existing online system (onlineMCA) as the control. This setup allows us to directly measure whether OneSearch can outperform the full production pipeline it is intended to replace, which is the most meaningful comparison for assessing real-world effectiveness. Furthermore, there is no publicly available e-commerce search dataset in the community that is constructed from real user behavior logs and includes user profiles, search queries, historical behavior, and interactions with each product, because this involves user privacy and business secrets. We plans to open source our evaluation dataset and core codes under the company's limitation, in order to fulfill ICLR's REPRODUCIBILITY STATEMENT pledges and provide a public benchmark for further exploration in academia.
>
> Thank you again for your careful review. If you have any further questions, please feel free to ask, and we will provide you with a thorough and detailed answer.

---

### Author Response · Authors · 2025-11-24
**General reply to reviewer comments**

Dear Reviewers

We would like to thank you for the time taken to carefully assess our paper and provide valuable feedback. We are very happy to hear your positive assessment of this paper. Based on your comments, we have uploaded a revised version of the paper in which we have incorporated all your comments and concerns. Below, we provide answers to each of your comments and questions individually. We hope that these replies clarify all your questions. Please do not hestiate to let us know in case any of our answers require further clarification.

---

### Note · Authors · 2026-01-26

I have read and agree with the venue's withdrawal policy on behalf of myself and my co-authors.

---

### Meta-Review · Area_Chair_HpkD · 2025-12-10

**Summary:**

I am under the impression that there are too many issues with the original submission in terms of novelty and importance. This may be caused by presenting a "system" with many features rather than a novel technical solution for a dedicated particular problem. A "system" has naturally many components and the limited space of a conference submission may not allow to go into too many details. Nevertheless, none of the reviewers is really enthusiastic about the paper and individiual scores include many "poor" ratings, indicating that the paper needs a major revision before publication would be OK. My conclusion is to reject the paper at the current stage.

**Reviewer Concerns:**

The main issue raised by the reviewers appears to be novelty. They also raised many other concerns about the writeup and presentation in general, choice of data sets, and reproducibility that probably have not been addressed sufficiently. For example, there seems to be a plan to publish data at some point but the text is rather vague and there is no timeline (e.g., upon acceptance, in a year from now...) for doing this. Instead, the authors highlight the industrial application and the corporate data. Nevertheless, this is an academic conference and we need to make sure that interested folks can assess the value of the publication on their own by repeating and reproducing the results so there is no good answer to this.

**Reviewer Scores:**

Dedicated questions (e.g., scalability, discussion of results, etc.) have certainly been addressed by the authors in the rebuttal but it is difficult to argue about novelty, particularly when the initial writeup is difficult to understand. Given the assessment above, I doubt that the rebuttal would have had a significant change in how the reviewers rate the submission.

---

### Decision · Program_Chairs · 2026-01-26

Reject